# Exact Privacy Guarantees for Markov Chain Implementations of the Exponential Mechanism with Artificial Atoms

**Jeremy Seeman**
Department of Statistics
Penn State University
State College, PA 16803
jhs5496@psu.edu

**Matthew Reimherr**
Department of Statistics
Penn State University
State College, PA 16803
mlr36@psu.edu

**Aleksandra Slavković**
Department of Statistics
Penn State University
State College, PA 16803
abs12@psu.edu

## Abstract

Implementations of the exponential mechanism in differential privacy often require sampling from intractable distributions. When approximate procedures like Markov chain Monte Carlo (MCMC) are used, the end result incurs costs to both privacy and accuracy. Existing work has examined these effects asymptotically, but implementable finite sample results are needed in practice so that users can specify privacy budgets in advance and implement samplers with exact privacy guarantees. In this paper, we use tools from ergodic theory and perfect simulation to design exact finite runtime sampling algorithms for the exponential mechanism by introducing an intermediate modified target distribution using artificial atoms. We propose an additional modification of this sampling algorithm that maintains its $\epsilon$-DP guarantee and has improved runtime at the cost of some utility. We then compare these methods in scenarios where we can explicitly calculate a $\delta$ cost (as in $(\epsilon, \delta)$-DP) incurred when using standard MCMC techniques. Much as there is a well known trade-off between privacy and utility, we demonstrate that there is also a trade-off between privacy guarantees and runtime.

## 1 Introduction

### 1.1 Problem setup

The exponential mechanism [17] is one of the workhorses of $\epsilon$-differential privacy ($\epsilon$-DP). Many common DP mechanisms, such as the Laplace Mechanism [7], K-norm and K-norm gradient mechanisms [22, 4], pMSE mechanism [24], and even posterior sampling under certain regularity conditions [26], can all be viewed as instances of the exponential mechanism. In fact, any mechanism can be equivalently expressed as an instance of the exponential mechanism depending on the choice of loss function and base measure [2]. However, the output of the exponential mechanism often has an intractable distribution, making it difficult to sample from in practice. This necessitates using a sampling algorithm such as Markov chain Monte Carlo (MCMC) to computationally generate an approximate sample from the distribution. However, methods like these only approximate the target distribution asymptotically, meaning that the resulting sample need not satisfy the originally prescribed privacy guarantees.

In practice, analyzing the privacy cost incurred due to sampling is a messy problem. Existing approaches rely on analyzing convergence rates of distances between the approximating distribution and the target distribution. While these approaches have produced nice theoretical results, the analyses are usually asymptotic and lack closed form expressions for constants required to determine finite-sample incurred costs. However, for uniformly ergodic chains that admit atoms in their state

35th Conference on Neural Information Processing Systems (NeurIPS 2021).

space, these sampling procedures are not only more directly quantifiable but also admit finite runtime algorithms for exact simulation from the target distribution. This suggests that modifying the target distribution of the exponential mechanism to include an artificial atom may eliminate the privacy cost of implementing the sampling algorithm; such modifications are the focus of this paper.

## 1.2 Contributions

Let $f_X(y)$ be the output density of the exponential mechanism with confidential data $X$ and output space $y \in \mathcal{Y}$ (we will fully define this notation in the methods section). In this paper, we consider sampling from $f_X$ by first sampling from a modified target density as a mixture of the form $k \in [0, 1]$:

$$\tilde{f}_X(y) = (1 - k)f_X(y) + kg_X(y).$$

In particular, we introduce two different methods which we will give names and briefly describe:

- `ConfAtomPerfect` $g_X(y)$ is a point mass at the confidential data output, and the target distribution is $\tilde{f}_X(y)$ conditioned on NOT releasing the confidential data, yielding an exact draw from $f_X$.

- `RandomAtomPerfect` $g_X(y)$ is the mass function for an implementation of the discrete exponential mechanism over a finite test point space.

Our contributions are as follows:

1. We use classical methods from MCMC theory to derive exact sampling algorithms for instances of the exponential mechanism with $(\epsilon, \delta)$-DP guarantees.

2. We propose an exact finite-runtime algorithm (`ConfAtomPerfect`) for implementing the exponential mechanism with $\epsilon$-DP guarantees using atomic regeneration.

3. We derive two modifications of the previous algorithm (i.e., `RandomAtomPerfect` and `ConfAtomAndRuntimePerfect`) that satisfy $\epsilon$-DP. The first modifies the target distribution and does not resample to sample from $f_X$, and the second allows the implementation to be independent of runtime, offering even stronger privacy guarantees.

4. We compare the proposed methods for a few explicit examples to demonstrate a new three way trade-off between, privacy, utility, and runtime.

## 1.3 Related literature

Results from MCMC theory have addressed privacy loss due to approximation by analyzing convergence rates of total variational distance [18] and Rényi divergence [12]. In the latter paper, the authors argue against characterizing the distance between the MCMC sampled distribution and the target distribution in terms of total variation because it lacks the privacy-preserving compositional properties of Rényi-DP, allowing for the derivation of convergence results with better asymptotic dependence on the dimension of the output space. Our work differs in a few key directions. First, we require weaker assumptions about the loss function. In particular, $V$-geometric ergodicity only requires sub-exponential tails, which relaxes strict convexity and/or Lipschitz assumptions about the loss function. For our perfect sampler, we only require uniform ergodicity of the original chain, which is often achieved by virtue of bounding the database space. Second, we consider simpler classes of base MCMC algorithms that may be sufficient for a wide class of problems where the output dimension is smaller than the input dimension. Third, our work derives algorithms with exact implementation steps and exact privacy guarantees. Previous results state their results in terms of asymptotic sample complexities for the Langevin dynamics step size and run time, neither of which are immediately helpful for practitioners who want to use these algorithms and calculate their provable guarantees.

## 2 Methods

### 2.1 Notation

We recall some definitions and theorems here:

**Definition 1** (Differential Privacy [7]). *Let $\mathcal{M} \triangleq \{\mu_X \mid X \in \mathcal{X}^n\}$ be a collection of probability measures over a common measurable space $(\mathcal{Y}, \mathcal{F})$ indexed by elements of $\mathcal{X}^n$. The mechanism $\mathcal{M}$ satisfies $(\epsilon, \delta)$-DP if, for all $B \in \mathcal{F}$ and $X, X' \in \mathcal{X}^n$ differing on one entry:*

$$\mu_X(B) \leq e^\epsilon \mu_{X'}(B) + \delta.$$

**Definition 2** (Exponential Mechanism [17]). *Let $L_X : \mathcal{X}^n \times \mathcal{Y} \mapsto [0, \infty]$ be a measurable loss function for each $X \in \mathcal{X}^n$. Suppose, for all adjacent $X, X' \in \mathcal{X}^n$ and $y \in \mathcal{Y}$:*

$$|L_X(y) - L_{X'}(y)| \leq \Delta_L < \infty.$$

*The exponential mechanism is the $(\epsilon, 0)$-DP mechanism defined by the collection of measures each with density:*

$$f_X(y) \propto \exp\left(-\frac{\epsilon L_X(y)}{2\Delta_L}\right),$$

*with respect to a common base measure $\nu(y)$ over $(\mathcal{Y}, \mathcal{F})$.*

Frequently, $\nu$ is taken to be the Lebesgue measure over a subset of $\mathbb{R}^d$. Throughout the paper, let $\{\mu_{m,X}\}_{m=1}^\infty$ be a sequence of probability measures such that $\mu_{m,X}$ converges in distribution to $\mu_X$ for all $X \in \mathcal{X}^n$. We assume the chain is defined by the transition kernel $\Pi_X : \mathcal{Y} \times \mathcal{F} \mapsto [0, 1]$ and the starting value $y_0 \in \mathcal{Y}$. Throughout the paper, we use the concept of minorization:

**Definition 3** (Minorization Condition). *Suppose there exists a function $s : \mathcal{Y} \mapsto [0, 1]$ and probability measure $\nu$ on $(\mathcal{Y}, \mathcal{F})$ such that for all $A \in \mathcal{F}$ and $y \in \mathcal{Y}$:*

$$\Pi_X(y, A) \geq s(y)\nu(A).$$

*We call these the minorization function and measure, respectively. Define the associated remainder kernel $R_{\nu,s} : \mathcal{Y} \times \mathcal{F} \mapsto [0, 1]$ such that:*

$$\Pi_X(y, A) = s(y)\nu(A) + (1 - s(y))R_{\nu,s}(y, A).$$

Our results rely on the existence of minorization functions and measures for our output distribution $\mu_X$. This assumption will give us uniform ergodicity properties later on. Note that a limitation of our approach is that we cannot apply these methods to exponential mechanisms with unbounded output spaces such as $\mathbb{R}^k$, but it does work for compact sets of $\mathbb{R}^k$. In practice, this limitation is not a major barrier to generalizability, as the the output space is often intentionally bounded to control the sensitivity of the loss function, although there are exceptions (e.g., [24]).

The following result shows that MCMC sampling from the exponential mechanism induces a $\delta_\alpha$ penalty as a function of the upper bound on the total variation distance $\alpha$:

**Theorem 1** ([18]). *A mechanism $\mathcal{M}_m \triangleq \{\mu_{m,X} \mid x \in \mathcal{X}\}$ approximating the exponential mechanism, $\mathcal{M}$, at time $m \geq \tau(\alpha)$ is $(\epsilon, \delta_\alpha)$-DP where $\delta_\alpha \triangleq \alpha(1 + e^\epsilon)$ if*

$$\tau(\alpha) \triangleq \sup_{X \in \mathcal{X}^n} \inf\{t \geq 0 \mid \|\mu_{t,X} - \mu_X\|_{\mathrm{TV}} \leq \alpha\}.$$

Note that the upper bound on the $\delta$ cost incurred above is specific to the sequence's properties (for example, transition probabilities and starting locations for MCMC). However, the upper bound is not specific to the confidential data $X$. Such generality is necessary to preserve $(\epsilon, \delta)$-DP, as the mixing time can vary based on the confidential data. This means to ensure privacy, one must find realizations of $X$ that are "least favorable" to ensure privacy in the worst-case scenario.

One common heuristic here is to choose $\delta \leq 1/n$, so that the relative privacy risk violation affects, in the worst case scenario, all information about one individual in the dataset. One common "fast-mixing" regime for these chains is a geometric rate, where the total variation distance is $O(r^m)$ for some $r \in (0, 1)$:

**Definition 4** ($V$-geometric ergodicity [1]). *For a Markov chain where $\mu_{m,X} \to \mu_X$ in distribution as $m \to \infty$, we say the chain is $V$-geometrically ergodic if there exists constants $C \in \mathbb{R}^+$, $r \in (0, 1)$, and a function $V : \mathcal{Y} \mapsto [1, \infty)$ such that for all $m \in \mathbb{N}$ and starting points $y_0 \in \mathbb{Y}$:*

$$\|\mu_{m,X} - \mu_X\|_{TV} \leq \|\mu_{m,X} - \mu_X\|_V \leq CV(y_0)r^{-m}. \tag{1}$$

*When $V$ is a constant function, we say the chain is uniformly ergodic.*

Many common algorithms, such as Metropolis-Hastings [1] and common variants such as Hybrid Monte Carlo [23] and Hamiltonian Monte Carlo [16], are geometrically ergodic under mild regularity conditions. For these algorithms, it immediately follows that for any specific realization $X \in \mathcal{X}^n$, if we choose $m = \Omega(\log(n))$:

$$\|\mu_{m,X} - \mu_X\|_{TV} \leq CV(y_0)r^{\log(n)} \implies \delta_\alpha = O(1/n). \tag{2}$$

Therefore, up to a constant, for a fast mixing sampling algorithm we require at least order $\log(n)$ samples to achieve a negligible $\delta$ for an empirical privacy guarantee. However, there are two complications to formalizing this idea. First, the constant of proportionality for the rate depends on $\epsilon$, $r$, $V$, and $C$. Second, the mixing time depends on $X$, and the privacy guarantee depends on fast mixing for all possible $X \in \mathcal{X}^n$.

## 2.2 Properties of Markov chains whose state spaces admit atoms

The complications above can be alleviated when the target distribution admits an atom, specifically a proper accessible atom with respect to the transition kernel $\Pi_X$:

**Definition 5** (Proper Accessible Atom). *A set $\gamma \in \mathcal{F}$ is a proper accessible atom if:*

1. *(Regeneration measure) There exists a measure $\mu_{\mathrm{regen}}$ on $(\mathcal{Y}, \mathcal{F})$ such that:*

$$\Pi_X(y, A) = \mu_{\mathrm{regen}}(A) \qquad \forall y \in \gamma, A \in \mathcal{F}.$$

2. *(Accessibility criteria)*

$$\Pi_X(y, \gamma) > 0 \quad \forall y \in \mathcal{Y}.$$

3. *(Recurrence criteria) the chain returns infinitely often to $\gamma$ with probability 1.*

When the state space admits an atom, the constants in Equation 1 have closed form expressions (see [1, Theorem 4.1] for an example). Furthermore, Markov chains that admit atoms have a special split chain representation [20], where the state space is extended with a binary indicator indicating presence in the atom (i.e., $\mathcal{Y} \times \{0, 1\}$). For this split chain, let $\tau$ be the time it takes for the split Markov chain to return to said proper accessible atom after leaving it. Hobert et al. [13] show that the stationary distribution admits an infinite mixture form:

$$\mu_X(A) = \sum_{m=1}^{\infty} \frac{\mathbb{P}(\tau \geq m)}{\mathbb{E}[\tau]} \mathbb{P}(Y_m \in A \mid \tau \geq m).$$

If one can sample from the mixture components as well as the conditional distributions feasibly (i.e., in finite expected time), then one can sample from $\mu_X$. Following [15], the authors focus on the singleton case when $\gamma = \{a\}$ for some $a \in \mathcal{Y}$. Define:

$$p(y) \triangleq \Pi_X(y, \{a\}).$$

If there exists $\beta > 0$ such that $\inf_{y \in \mathcal{Y}} p(y) > \beta > 0$, then wen can use Algorithm 1 to produce an exact sample from the target distribution.

Because many target distributions do not admit the atom required, Brockwell [6] proposes a way to introduce an artificial atom into a new stationary density $\tilde{f}_X$. In our setting, we propose introducing an atom at the confidential response mixed with the $\epsilon$-DP stationary distribution with proportions $k$ and $1 - k$, respectively, for some $k \in (0, 1)$:

$$\tilde{f}_X(y) \triangleq (1 - k)f_X(y) + k\mathbb{1}_{\{y=a\}},$$

with respect to a new base measure:

$$\tilde{\nu} \triangleq (1 - k)\nu + k\xi_a,$$

where $\nu$ is the original base measure and $\xi_a$ is the Dirac delta function representing a point mass at the confidential data. In particular, if the Metropolis-Hastings kernel has symmetric proposal density $q_X(y, y')$, then one can construct a new Metropolis-Hastings transition kernel $\tilde{\Pi}$ with symmetric proposal density:

$$\tilde{q}(y, y') = \frac{1}{2} \left[ q_X(y, y') + \mathbb{1}_{\{y'=a\}} \right].$$

---

**Algorithm 1:** Exact implementation for sampling from a singleton atom mixture [15]

---

**Result:** $Y \sim \tilde{\mu}_X$
**Input:** Transition kernel $\Pi_X$, singleton atom $a \in \mathcal{Y}$, minorization $\beta > 0$
Sample $M \sim \text{Geometric}(\beta)$, set $Y_1 = a$ ;
**for** $m = 2$ *to* $M$ **do**
  Using the Bernoulli factory algorithm [14], sample:
$$Z_m \sim \text{Bernoulli}\left(\frac{1 - \tilde{\Pi}_X(Y_{m-1}, \{a\})}{1 - \beta}\right)$$

  **if** $Z_m = 1$ **then**
    Sample $Y_m \sim R_{\xi_a, \tilde{\Pi}_X(\cdot, \{a\})}(Y_{m-1}, \cdot)$ using rejection:
      1. Propose $Y_m^* \sim \tilde{\Pi}_X(Y_{m-1}, \cdot)$.
      2. Accept if $Y_m^* \neq a$, else go back to 1.
  **else**
    $Y_m = a$
  **end**
  Release $Y_M$.
**end**

---

## 2.3 Extensions to private sampling

The framework established by [15] requires assumptions that are particularly amenable to privacy preservation. First, the minorization conditions needed to ensure finite runtime are often satisfied by design; most Markov chains are implemented over compact state spaces, where the compactness is enforced so that the exponential mechanism's loss function has bounded sensitivity. Second, we can choose to add an artificial atom at the confidential data, namely any point $a \in \mathcal{Y}$ such that $L_x(a) = 0$ without loss of generality (note that $a$ need not be unique). This yields our first main result in Theorem 2, and Algorithm 2.

**Theorem 2** (`ConfAtomPerfect`). *Suppose that $\mathcal{Y}$ is a compact subset of $\mathbb{R}^d$ and let $L_X$ be a loss function based on confidential data $x \in \mathcal{X}^n$ where $L_X(a) = 0$ for some $a \in \mathcal{Y}$. Suppose $\mu_X(\{a\}) = 0$. Let $Q$ be a Metropolis-Hastings transition kernel with symmetric proposals, and let $\tilde{Q}$ and $\tilde{q}$ be as defined above. Then:*

1. *There exists a constant $\beta > 0$ such that:*
$$\inf_{y \in \mathcal{Y}} \tilde{\Pi}_X(y, \{a\}) > \beta.$$

2. *There exists an algorithm (implemented by Algorithm 1) to sample from density $f_X$ that satisfies $\epsilon$-DP with expected number of total proposed samples $N_{\text{prop}}$:*
$$\mathbb{E}[N_{\text{prop}}] \leq \frac{48}{k^2(1-k)^2 \inf_{y \in \mathcal{Y}} p_{\text{Accept}}(y)},$$
   *where:*
$$p_{\text{Accept}}(y) \triangleq \int_{\mathcal{Y}} q_X(y, y') \min\left\{1, \frac{f_X(y')}{f_X(y)}\right\} \, d\nu(y').$$

*Proof.* (Sketch; see complete proof in appendix) By construction, the Metropolis-Hastings transition kernel is maximized at $a$ for the modified chain. Taking $\beta \triangleq \frac{k}{2}$ yields the first result.

For the second result, the runtime is split into four components:

$$\begin{cases} N_{\text{Outer}} \triangleq \text{Number of outer loops in Algorithm 1,} \\ N_{\text{Inner}} \triangleq \text{Number of rejection proposals to sample from } Y_m^* \sim R_{a,p}(Y_{m-1}, \cdot), \\ N_{\text{Bern}} \triangleq \text{Number of Bernoulli factory samples to select } Z_m \text{ in Algorithm 1,} \\ N_{\text{Nonatommic}} \triangleq \text{Number of samples from } \tilde{f}_X \text{ required to sample from } f_X. \end{cases}$$

In the complete proof, we show:

$$\begin{cases} \mathbb{E}[N_{\text{Outer}}] = \frac{2}{k}, \\ \mathbb{E}[N_{\text{Inner}}] \leq \left((1-k)\inf_{y \in \mathcal{Y}} p_{\text{Accept}}(y)\right)^{-1}, \\ \mathbb{E}[N_{\text{Bern}}] \leq \frac{24}{k}, \\ \mathbb{E}[N_{\text{Nonatommic}}] = (1-k)^{-1}. \end{cases}$$

This yields the desired bound. $\qquad\square$

---

**Algorithm 2:** `ConfAtomPerfect`: $\epsilon$-DP exact sample from exponential mechanism

---
**Result:** $Y \sim f_X$
**Input:** Sample space $\mathcal{Y}$ and loss function $L_x$ satisfying the conditions of Theorem 2
**while** TRUE **do**
  Sample $\tilde{Y} \sim \tilde{f}_x$ using Algorithm 1. **if** $\tilde{Y} \neq a$ **then**
   Release $\tilde{Y}$.
  **end**
**end**

---

We make a couple remarks about this result. First, we note that the bound in the runtime can depend on the confidential data and does not require us to take the infimum over all possible data sets. While this is a nice property, it also assumes that the runtime of the algorithm is not a possible source of side channel information (more on this in the discussion). Second, the result above only uses symmetric proposals so that the criteria for transitioning to the confidential data does not depend on the proposal distribution. However, this result can easily be extended to arbitrary proposals in cases where the minorizing constant can still be calculated. Third, note that we do not include the case where $\mu_X(\{a\}) > 0$, as this corresponds to an exact implementation of the algorithm derived in [15]. Finally, we anticipate the expected runtime bound above is very loose; the bounds $N_{\text{Bern}}$ and $N_{\text{Inner}}$ are generic, and can be much lower for specific implementations. Moreover, $N_{\text{prop}}$ is loosely bounded above by the product of the terms in the proof sketch, but in practice not every branch of Algorithm 1 is traversed each time.

Algorithm 2 relies on multiple passes through Algorithm 1 in order to sample from the original target distribution; however, upon failure to do so, we could alternatively sample from a discrete approximation of the exponential mechanism target distribution. This yields a less computationally expensive method, avoiding the additional $(1-k)^{-1}$ factor in implementing the mechanism with pure $\epsilon$-DP, at the expense of some loss in utility due to the discrete approximation.

**Corollary 1** (`RandomAtomPerfect`). *In the same setup as Theorem 2, let $\{y^{(l)}\}_{l=1}^{\ell} \subset \mathcal{Y}$ and let $g_X$ be the mass density of the exponential mechanism implemented on $\{y^{(l)}\}_{l=1}^{\ell}$. Then there exists a sampling algorithm (implemented by Algorithm 3) with target density:*

$$\tilde{f}_X(y) = (1-k)f_X(y) + kg_X(y), \tag{3}$$

*with respect to the mixture measure:*

$$\tilde{\eta} = (1-k)\nu + \frac{k}{\ell}\sum_{l=1}^{\ell}\xi_l,$$

*with expected number of total proposed samples $N_{\text{prop}}$:*

$$\mathbb{E}[N_{\text{prop}}] \leq \frac{48}{k^2(1-k)\inf_{y \in \mathcal{Y}} p_{\text{Accept}}(y)},$$

*And utility:*

$$\mu_X(S_\varepsilon) \leq \frac{1-k}{\nu(S_\varepsilon/2)}\exp\left(-\frac{\epsilon\varepsilon}{4\Delta_L}\right) + k\exp\left(-\frac{\epsilon}{2\Delta_L}\left(\varepsilon - \frac{2\Delta_L}{\epsilon}\log(\ell)\right)\right),$$

*Where $S_\varepsilon \triangleq \{y \in \mathcal{Y} \mid L_X(y) \geq \varepsilon\}$.*

---

**Algorithm 3:** `RandomAtomPerfect`: $\epsilon$-DP sample from mixture of discrete and continuous exponential mechanisms

---

**Result:** $Y \sim \tilde{f}_X$ as in Equation 3
**Input:** Sample space $\mathcal{Y}$ and loss function $L_x$ satisfying the conditions of Corollary 1
Sample $\tilde{Y}$ using Algorithm 1.
**if** $\tilde{Y} \neq a$ **then**
  | Release $\tilde{Y}$.
**else**
  | Release $Y \sim g_X$
**end**

---

One potential issue with both algorithms 2 and 3 is that $N_{\text{Inner}}$ contains confidential information about $X$, presenting itself as a potential form of side leakage. However, following [3], we can exploit the memorylessness property of the Geometric distribution to release a result whose runtime is independent of the data (at a cost due to runtime). This is implemented in Corollary 2.

**Corollary 2** (`ConfAtomAndRuntimePerfect`). *In the same setting as Theorem 2, there exists a modification of Algorithm 2 where:*

1. *The total number of proposed samples (either real or partially artificial) $N_{\text{prop}}$ is independent of the data (i.e., satisfies $0$-DP).*

2. *The expected runtime of the algorithm is upper bounded by:*

$$\mathbb{E}[N_{\text{prop}}] \leq \frac{48}{k^2(1-k)^2\eta},$$

*Where:*

$$\eta \triangleq \frac{1}{2} \inf_{x \in \mathcal{X}^n} \inf_{y \in \mathcal{Y}} p_{\text{Accept}}(y)$$

## 3 Examples

### 3.1 $d$-dimensional mean with $L_1$ loss: comparing $\epsilon$-DP and $(\epsilon, \delta)$-DP guarantees

As a first example, let $\mathcal{Y} = [0,1]^d$, $L_X(y) \triangleq \left\| y - \overline{X} \right\|_1$, and $\nu$ be a uniform measure over $\mathcal{Y}$. This is analogous to the multivariate Laplace mechanism but with bounded output space matching the bounded inputs we expect from $\mathcal{X}$. Note that although we can implement exact samplers in this scenario, the goal of this example is to illustrate the effect of proposal choice on runtime and privacy costs using MCMC versus using Algorithm 2. Derivation of all example results are in the appendix.

We consider two different proposals for MCMC. For the first algorithm, we derive a Metropolis-Hastings Markov chain constructed with proposal independently drawn uniformly over $[0,1]^d$. The slowest-mixing chain is easily identifiable using results from [1]:

$$\|\mu_{X,m} - \mu_X\|_{TV} \leq (1 - \beta_{\text{MCMC,unif}})^m, \tag{4}$$

where,

$$\beta_{\text{MCMC,Unif}} \triangleq \left( \frac{2d}{\epsilon n}(1 - e^{-\epsilon n/2d}) \right)^d.$$

Derivation of this result is in the appendix. This allows us to immediately construct $(\epsilon, \delta)$-DP implementations of the exponential mechanism. Note that, using the main result in [25], Equation 4 is constant-sharp and cannot be improved.

Similarly, we can use Laplace proposals of the form:

$$q_X(y, y') \propto \exp\left( -\alpha \|y - y'\|_{L_1} \right),$$

for which we have a new rate of convergence given analogously by:

$$\beta_{\text{MCMC,Lap}} \triangleq (2\alpha)^d \exp\left( -\left( \alpha d + \frac{\epsilon n}{2} \right) \right) \left( \frac{1}{\alpha}(1 - e^{-\alpha}) \right)^d.$$

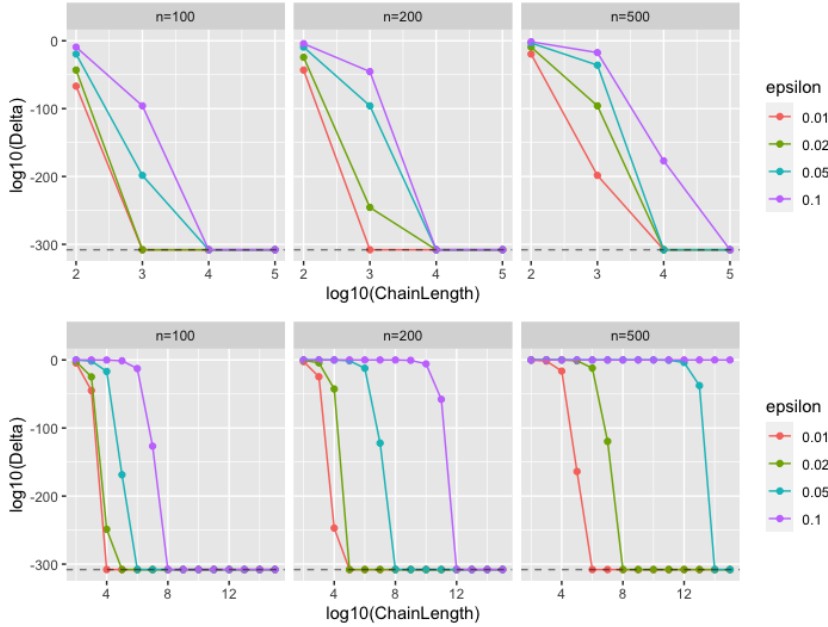

Figure 1: Delta costs incurred due to MCMC implementation with uniform independent proposals and symmetric Laplace proposals, top and bottom respectively. Dashed line corresponds to smallest 64-bit double floating point precision; note this does NOT mean the resulting algorithm is $\epsilon$-DP, but instead that the $\delta$ incurred is smaller than the smallest floating-point value reportable by the machine.

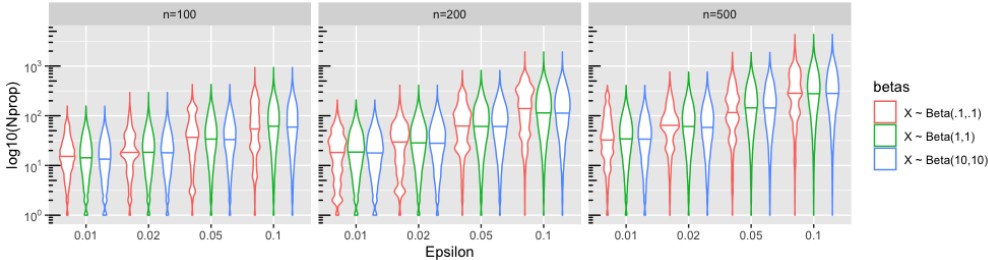

Figure 2: Distribution of realized $N_{\mathrm{prop}}$ for Example 1 with symmetric Laplace proposals at $\alpha = n\epsilon/2$, $d = 1$, and different distributions on $\overline{X}$ (line represents median). As expected, run-time increases as both $n$ and $\epsilon$ increases.

The runtime for Algorithm 2 depends on the confidential data. To demonstrate this, we simulate $\overline{X}$ from three different Beta distributions: one uniform ($\overline{X} \sim \mathrm{Beta}(1,1)$), one with unimodal concentration ($\overline{X} \sim \mathrm{Beta}(10,10)$), and one with concentration at the boundary ($\overline{X} \sim \mathrm{Beta}(.1,.1)$) and compare the distribution of $N_{\mathrm{prop}}$ across the cases in Figure 2. All simulations are implemented in R [21] and figures were produced using ggplot2 [27] under GNU GPL v3.

The runtime results presented in figures 1 and 2 illustrate the phenomena in the discussion for Theorem 2. For the MCMC implementations in Figure 1, even though we expect the Laplace proposals to mix faster in general, the worst case mixing time is actually better for the independent uniform proposals. For example, at $n = 100$ and $\epsilon = .01$, we require a Laplace proposal chain 10 times longer than the equivalent uniform proposal chain to achieve a computationally negligible $\delta$. Alternatively for the perfect sampler in Figure 2, the runtime for the perfect sampler decreases when the data is more likely to be concentrated at the center of the output space (as in $\mathrm{Beta}(10,10)$) as opposed to the boundary of the space (as in $\mathrm{Beta}(.1,.1)$).

### 3.2 Ridge regression K-norm gradient mechanism: comparing confidential and random atoms

Next, we consider Ridge regression [11]. We will use the K-norm gradient mechanism following [22], a variant of the original exponential mechanism whose noise is asymptotically negligible as $n \to \infty$. In the interest of keeping consistent notation, let:

$$\mathcal{Y} \triangleq \{y \in \mathbb{R}^p \mid \|y\|_1 \leq B\}.$$

Let $X \in [-1, 1]^{n \times p}$ and $Z \in [-1, 1]^n$. Define the original loss function $\ell_X(y)$:

$$\ell_X(y) = \frac{1}{2} \|Z - Xy\|_2^2 + \frac{\lambda}{2} \|y\|_2^2.$$

For any two $X, X'$ that differ on one element and all $y \in \mathcal{Y}$:

$$\|\nabla \ell_X(y) - \nabla \ell_{X'}(y)\|_1 \leq 2(1 + B) \sup_{x \in [0,1]^p} \|x\|_1 \leq 2(1 + B)p$$

This yields a final mechanism given by:

$$f_X(y) \propto \exp\left(-\frac{\epsilon}{4(1 + B)p} \left\|X^T(Xy - Z) + \lambda y\right\|_1\right) \mathbb{1}_{\{\|y\|_1 \leq B\}}. \tag{5}$$

Because of similarities between functional forms, the exact sampling procedures will behave the same as the analogues in the previous example. In particular, note that the worst-case runtime calculation does not depend on the integration constant in Equation 5 when the proposals $q_X$ are symmetric, since:

$$p_{\text{Accept}}(y) = \int_{\|y\|_1 \leq B} q_X(y, y') \min\left\{1, \frac{\exp\left(-\frac{\epsilon}{4(1+B)p} \left\|X^T(Xy' - Z) + \lambda y'\right\|_1\right)}{\exp\left(-\frac{\epsilon}{4(1+B)p} \left\|X^T(Xy - Z) + \lambda y\right\|_1\right)}\right\} d\nu(y').$$

This property is not shared by the MCMC estimator, demonstrating an advantage of this procedure.

Our main goal is to compare the utilities of Algorithms 2 and 3, meaning we need to quantify the utility cost of discretizing the output space with some probability $k$. To analyze this, we simulate data from a Ridge regression model with $p = 5$ and randomly sample $\ell$ points from the $L_1$ ball of radius $B = 1$ on which to implement the discrete exponential mechanism. By varying $\ell$ and $\epsilon$ and repeating the experiment $N_{\text{exp}} = 1000$ times, we calculate:

$$\text{Err}(\epsilon, \ell, \varepsilon) \triangleq Q_{.05}\{P(L_X(y)^{(n_{\text{exp}}, \epsilon, \ell)} \geq \varepsilon)\}.$$

where $Q_{.05}$ is the 5% quantile across experiment replications. This captures with approximately 95% confidence a kind of "worst case" utility bounds by choosing a particular discretization at random for our Ridge regression. The results of the experiment are in Figure 3 with full simulation details in the appendix. Since the discrete exponential mechanism utility is a Monte Carlo estimate of the continuous exponential mechanism utility, we see as expected that the utility increases to the continuous upper bound as $\ell \to \infty$. For our particular example, this suggests that we could choose $\ell$ large enough that the utility of the discrete mechanism is close to that of the original exponential mechanism with high probability. However, further investigation is needed to understand these effects in higher dimensions or for more complicated loss functions.

## 4 Discussion

We proposed some techniques for implementing the exponential mechanism with exact privacy guarantees and some pros and cons of the various approaches above. We hope that techniques like these can be considered when practitioners are either unwilling to incur a $\delta$ cost or are otherwise worried about the leakage of sensitive information due to disclosure of implementation details. Additionally, our corollaries and simulations demonstrate new ways of trading off privacy, utility, and now run-time, whereas traditionally only the trade-off between the first two was considered. This can help practitioners quantitatively negotiate the realized costs of implementing more generic DP algorithms.

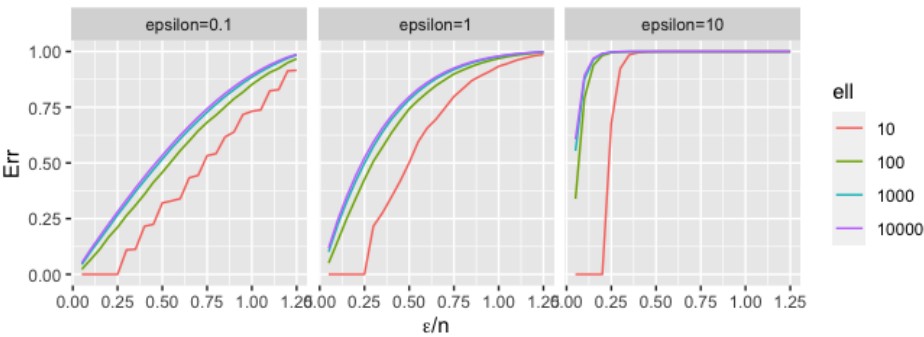

Figure 3: Utility of the discrete exponential mechanism using $\ell$ points sampled uniformly from the support of the distribution in Equation 5.

## 4.1 Limitations

As with any real-valued private computation, we must be careful about the complications of floating-point arithmetic and other side channel vulnerabilities [19]. Just because a machine cannot report a $\delta$ cost below its floating point precision does not mean the system is invulnerable. In particular, runtime issues pose a practical threat to privacy preservation when using these mechanisms; longer runtimes tend to correspond to rarer events, leading to indirect leakage. This implies our methods are best used for non-interactive data analyses, in which runtime information leakage is a lesser concern. Future work could address methods of masking or privatizing the total runtime in a way that masks how much computation time went into sampling from the target distribution.

Additionally, we have restricted our examples to relatively well-behaved distributions that allow us to analytically characterize the constants needed for exact sampling. Numerical analysis of these constants are not the primary focus of the literature on perfect sampling, but understanding additional practical applications to estimate or otherwise bound these constants would be worth investigating. Additionally, the methods we propose suffer from curse of dimensionality problems and may become prohibitively expensive for large $d$ as noted in [12]. Still, many common problems fall into the regime where $d \ll n$ and $\mathcal{X}^n$ is not discrete; this is where we believe our work is most applicable.

Finally, we reiterate that uniform ergodicity of the original chains is not only sufficient but often necessary for the existence of perfect sampling methods with finite expected runtime [15]. However, the minorization constant suffers from curse of dimensionality effects, meaning the runtime will often suffer for high-dimensional results. Feasible and efficient algorithms for perfect sampling in high-dimensional settings remains an open challenge in mathematics.

## 4.2 Future research

First, understanding the utility trade-off between Algorithms 2 and 3 requires a deeper understanding of generic utility for the exponential mechanism. Although there has been some of work on the asymptotic utility of the exponential mechanism such as [2, 10], surprisingly little work has been done on exact finite sample guarantees. In the special case when the loss function is convex, the exponential mechanism's target density is log-concave, yielding many nice properties in terms of information concentration [5]. These should be further investigated to determine when modifications to the target distribution like those proposed here are prohibitive to utility in realized terms.

Additionally, the upper bound provided in Theorem 1 may be loose in practice, and alternative convergence diagnostics besides total variation distance may be used to better quantify the privacy guarantees up to an approximation (see [9] for a comparison of some Monte Carlo error estimation approaches). We expect there to be cases where Langevin dynamics or other sampling techniques with desirable mixing properties in practice could quickly produce mathematically negligible $\delta$ values; the only open question is turning this heuristic into a quantifiable guarantee.

## 5 Acknowledgements

This work is supported by NSF SES-1853209. Thanks to Jordan Awan and Alexei Novikov for helpful discussions about this topic. We declare no conflicts of interest.

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
