## B Proofs

### B.1 Proof of Theorem 2

First note that if there are multiple $a \in \mathcal{Y}$ for which $L_X(a) = 0$, we can sample $a \sim \mathrm{Unif}(\{a \in \mathcal{Y} \mid L_X(a) = 0\})$. Note that if $\mu_X(\{a \in \mathcal{Y} \mid L_X(a) = 0\}) > 0$, then we can directly apply the main

result from [15]. Therefore we instead focus on the case where $\mu_X(\{a \in \mathcal{Y} \mid L_X(a) = 0\}) = 0$ in agreement with our assumption.

Next, the Metropolis-Hastings transition kernel is defined by:

$$\tilde{\Pi}_X(y, A) = \int_A \tilde{q}_X(y, y') \min\left\{1, \frac{\tilde{f}_X(y')}{\tilde{f}_X(y)}\right\} d\tilde{\nu}(y') +$$

$$\mathbb{1}_{\{y \in A\}} \left[1 - \int_{\mathcal{Y}} \tilde{q}_X(y, y') \min\left\{1, \frac{\tilde{f}_X(y')}{\tilde{f}_X(y)}\right\} d\tilde{\nu}(y')\right],$$

Where:

$$\tilde{q}_X(y, y') \min\left\{1, \frac{\tilde{f}_X(y')}{\tilde{f}_X(y)}\right\} = \frac{1}{2}\left[\mathbb{1}_{\{y'=a\}} + q_X(y, y')\right] \min\left\{1, \frac{(1-k)f_X(y') + k\mathbb{1}_{\{y'=a\}}}{(1-k)f_X(y) + k\mathbb{1}_{\{y=a\}}}\right\}$$

The density $\tilde{f}_X$ is maximized at $a$ by construction; note that this does not depend on the uniqueness of $a$. This implies:

$$\tilde{\Pi}_X(y, \{a\}) \geq \int_{\{a\}} \tilde{q}_X(y, y') \min\left\{1, \frac{(1-k)f_X(y') + k}{(1-k)f_X(y) + k\mathbb{1}_{\{y=a\}}}\right\} d\tilde{\nu}(y')$$

$$\geq \frac{k}{2} \triangleq \eta > 0$$

Therefore the first condition is met, and we can use Algorithm 1 to perform perfect sampling. Let $N_{\text{Outer}} \sim \text{Geometric}(\eta_1)$ be the length of this outer loop.

Next, we need to characterize $N_{\text{Bern}}$, the number of Bernoulli factory flips necessary to sample from the Bernoulli distribution in Algorithm 1. Using the proposed algorithm in [14] and the minorization term above:

$$\mathbb{E}[N_{\text{Bern}}] \leq \frac{12}{\eta} = \frac{24}{k}$$

Next, we need to calculate, in the worst possible case, how many inner loop samples $N_{\text{Inner}}$ are necessary to sample from the remainder in Algorithm 1:

$$\tilde{\Pi}_X(y, \mathcal{Y} \setminus \{a\}) = \int_{\mathcal{Y} \setminus \{a\}} \tilde{q}_X(y, y') \min\left\{1, \frac{\tilde{f}_X(y')}{\tilde{f}_X(y)}\right\} d\tilde{\nu}(y') +$$

$$\mathbb{1}_{\{y \in \mathcal{Y} \setminus \{a\}\}} \left[1 - \int_{\mathcal{Y}} \tilde{q}_X(y, y') \min\left\{1, \frac{\tilde{f}_X(y')}{\tilde{f}_X(y)}\right\} d\tilde{\nu}(y')\right]$$

$$\geq (1-k) \int_{\mathcal{Y}} q_X(y, y') \min\left\{1, \frac{f_X(y')}{f_X(y)}\right\} d\nu(y')$$

Define:

$$p_{\text{Accept}}(y) \triangleq \int_{\mathcal{Y}} q_X(y, y') \min\left\{1, \frac{f_X(y')}{f_X(y)}\right\} d\nu(y')$$

Then:

$$\Pi_X(y, \mathcal{Y} \setminus \{a\}) \geq (1-k) \inf_{y \in \mathcal{Y}} p_{\text{Accept}}(y)$$

Finally, let $N_{\text{Nonatomic}}$ be the number of runs of Algorithm 1 to yield a perfect sample from the unmodified target distribution 2. Then $N_{\text{Nonatomic}} \sim \text{Geometric}(1-k)$. Combining all the results, the total number of proposed samples of all kinds $N_{\text{Total}}$ is bounded above by:

$$\mathbb{E}[N_{\text{Total}}] \leq \mathbb{E}[N_{\text{Outer}}] \mathbb{E}[N_{\text{Bern}}] \mathbb{E}[N_{\text{Inner}}] \mathbb{E}[N_{\text{Nonatomic}}] \leq \frac{48}{k^2(1-k)^2 \inf_{y \in \mathcal{Y}} p_{\text{Accept}}(y)}$$

## B.2 Proof of Corollary 1

The expected runtime bound follows immediately from the proof of Theorem 2 above. For the utility, recall that for the original exponential mechanism [17, Lemma 7]:

$$\mu_X(S_\varepsilon) \leq \frac{1}{\nu(S_{\varepsilon/2})} \exp\left(-\frac{\epsilon\varepsilon}{4\Delta_L}\right)$$

For the mechanism as implemented over the discrete atoms $\{y^{(l)}\}_{l=1}^\ell$, [8, Corollary 3.12] show that:

$$\mathbb{P}\left(\|L_X(Y)\| \geq \varepsilon \mid Y \in \{y^{(l)}\}_{l=1}^\ell\right) \leq \exp\left(-\frac{\epsilon}{2\Delta_L}\left(\varepsilon - \frac{2\Delta_L}{\epsilon}\log(\ell)\right)\right)$$

The utility result follows immediately from conditioning on the two mixture components.

## B.3 Proof of Corollary 2

Using the same notation from the proof of the main theorem, the only modification necessary so that $N_{\mathrm{prop}}$ is 0-DP is that the data-dependent component, $N_{\mathrm{inner}}$ have a distribution independent of $X$. Using [3] Lemma 17, we can add geometric random noise to $N_{\mathrm{Inner}}$ for any iteration of the inner loop with probability depending on $X$. In particular, we assume an adversary knows a modified $\tilde{N}_{\mathrm{Inner}}$ where:

$$\tilde{N}_{\mathrm{Inner}} \triangleq N_{\mathrm{Inner}}Z + (1-Z)(N_{\mathrm{Inner}} + N_{\mathrm{Wait}})$$

where:

$$Z \sim \mathrm{Bernoulli}\left(\frac{\inf_{X\in\mathcal{X}^n}\inf_{y\in\mathcal{Y}}p_{\mathrm{Accept}}(y)}{\inf_{y\in\mathcal{Y}}p_{\mathrm{Accept}}(y)}\right), \quad N_{\mathrm{Wait}} \sim \mathrm{Geometric}\left(\inf_{X\in\mathcal{X}^n}\inf_{y\in\mathcal{Y}}p_{\mathrm{Accept}}(y)\right)$$

Then:

$$\mathbb{E}\left[\tilde{N}_{\mathrm{Inner}}\right] \leq \frac{2}{(1-k)\inf_{X\in\mathcal{X}^n}\inf_{y\in\mathcal{Y}}p_{\mathrm{Accept}}(y)}$$

The corollary result then follows from replacing $\mathbb{E}\left[N_{\mathrm{Inner}}\right]$ with $\mathbb{E}\left[\tilde{N}_{\mathrm{Inner}}\right]$ in the the proof for Theorem 2.

## B.4 Derivation for Example 1

$$f_X(y) = C_X \mathbb{1}_{\{y\in[0,1]^d\}} \exp\left(-\frac{\epsilon n}{2d}\|y - \overline{X}\|_1\right)$$

With integration constant:

$$C_X^{-1} \triangleq \int_{[0,1]^d} \exp\left(-\frac{\epsilon n}{2d}\|y - \overline{X}\|_1\right) d\nu(y)$$

Independent uniform proposal MCMC sampler:

$$\inf_{X\in\mathcal{X}^n}\inf_{y\in\mathcal{Y}}\frac{q_X(y)}{f_X(y)} = \left(\sup_{X\in\mathcal{X}^n}\sup_{y\in\mathcal{Y}}f_X(y)\right)^{-1}$$

$$= \left(\inf_{X\in\mathcal{X}^n}C_X^{-1}\right)^{-1}$$

$$= \prod_{j=1}^d \int_0^1 \exp\left(-\frac{\epsilon n}{2d}|y_j|\right) dy_j$$

$$= \left(\frac{2d}{\epsilon n}(1 - e^{-\epsilon n/2d})\right)^d \triangleq \beta_{\mathrm{MCMC,Unif}}$$

Laplace proposal MCMC sampler; first, let $Z \sim \mathrm{MVLaplace}(x, \alpha)$ for $x \in [0,1]^d$ Then using the previous result:

$$\mathbb{P}(Z \in [0,1]^d) \geq \left(\frac{1}{\alpha}(1 - e^{-\alpha})\right)^d.$$

Then:

$$Q_X(y, y') \geq Q_X(\overline{X}, y')$$

$$\geq (2\alpha)^d \exp\left(-\left(\alpha d + \frac{\epsilon n}{2}\right)\right)\left(\frac{1}{\alpha}(1 - e^{-\alpha})\right)^d \triangleq \beta_{\text{MCMC,Lap}}$$

Let $F(\cdot; b)$ be the CDF of the Laplace distribution with scale parameter $b$. Independent uniform proposal perfect sampler:

$$p_{\text{Accept}}(y) = \int_{\mathcal{Y}} q_X(y, y') \min\left\{1, \frac{f_X(y')}{f_X(y)}\right\} d\nu(y')$$

$$= \int_{[0,1]^d} \min\left\{1, \exp\left(-\frac{\epsilon n}{2d}\left(\|y' - \overline{X}\|_1 - \|y - \overline{X}\|_1\right)\right)\right\} d\nu(y')$$

$$\geq \int_{[0,1]^d} \exp\left(-\frac{\epsilon n}{2d}\left(\|y' - \overline{X}\|_1\right)\right) d\nu(y')$$

$$= \prod_{j=1}^d \int_0^1 \exp\left(-\frac{\epsilon n}{2d}|y'_j - \overline{X}_j|\right) dy'_j$$

$$= \prod_{j=1}^d \frac{\epsilon n}{4d}\left(F\left(1 - \overline{X}_j; \frac{\epsilon n}{2d}\right) - F\left(-\overline{X}_j; \frac{\epsilon n}{2d}\right)\right)$$

Laplace proposal perfect sampler:

$$p_{\text{Accept}}(y) = \int_{\mathcal{Y}} q_X(y, y') \min\left\{1, \frac{f_X(y')}{f_X(y)}\right\} d\nu(y')$$

$$= \int_{[0,1]^d} (2\alpha)^d \exp\left(-\alpha\|y - y'\|_1\right) \min\left\{1, \exp\left(-\frac{\epsilon n}{2d}\left(\|y' - \overline{X}\|_1 - \|y - \overline{X}\|_1\right)\right)\right\} d\nu(y')$$

$$\geq \int_{[0,1]^d} (2\alpha)^d \exp\left(-\left(\frac{\epsilon n}{2d} + \alpha\right)\left(\|y' - \overline{X}\|_1\right)\right) d\nu(y')$$

$$= (2\alpha)^d \prod_{j=1}^d \int_0^1 \exp\left(-\left(\frac{\epsilon n}{2d} + \alpha\right)|y'_j - \overline{X}_j|\right) dy'_j$$

$$= (2\alpha)^d \prod_{j=1}^d \left(\frac{\epsilon n}{4d} + \frac{\alpha}{2}\right)\left(F\left(1 - \overline{X}_j; \frac{\epsilon n}{2d} + \alpha\right) - F\left(-\overline{X}_j; \frac{\epsilon n}{2d} + \alpha\right)\right)$$

## B.5   Simulation specification for Example 2

Constants:

$$\begin{cases} n \triangleq = 100 \\ p \triangleq 5 \\ \beta \triangleq (.1, .2, -.3, 0, 0)^T \\ \lambda \triangleq 1 \end{cases}$$

Random variables:

$$\begin{cases} X_{ij} \sim \text{Beta}(5, 5) & i \in [n], j \in [p] \\ e_i \sim \text{Beta}(20, 20) & i \in [n] \\ Z_i \triangleq X_{i,\cdot}\beta + (2e_i - 1) & i \in [n] \end{cases}$$