# OpenReview forum: "Exact Privacy Guarantees for Markov Chain Implementations of the Exponential Mechanism with Artificial Atoms"
_NeurIPS.cc/2021/Conference — NeurIPS 2021 Poster_

### Official Review · Reviewer_DMjk · 2021-06-29

**Rating:** 6
**Confidence:** 4

**Summary:**

This paper explores the use of finite runtime samplers applied on exponential mechanism (EM). The paper presents two main contributions: (1) an algorithm to produce exact samples from EM under certain conditions, (2) an algorithm to alleviate computational cost of contribution (1) with a loss in utility. There are also experiments that demonstrates the utility tradeoff between (1) and (2).

**Limitations And Societal Impact:**

Yes

**Main Review:**

Sampling from EM using classical MCMC sampling techniques (such as Metropolis-Hastings) is in general hard, since it has been difficult to guarantee that the chain has actually converged to the target distribution which is mandatory for the privacy guarantee. This paper proposes a method(s) to assert the correctness of the samples by using established techniques from MCMC literature. The main workhorse of this paper is sampling with atoms, that are sort of checkpoints in sampling domain. Samples between visits in the atom can be treated as samples from the target distribution and therefore guarantee the exactness of the method.

Augmenting the original sampling problem (sampling from EM) with a sampling from an atom component, authors are able to provide $\epsilon$-DP for loss functions that have unique zeros. I think this approach is rather novel and compared to the previous work by Ganesh and Talwar [1], the present work quantifies the $\epsilon$-DP instead of pushing the uncertainty in sampling to the $\delta$ privacy parameter. I also like the proposed algorithm in it's simplicity (it's a basic rejection sampler).

However, there are several weaknessess in the approach that make it less appealing. First, authors present the math for only a few very simple transition kernels. I doubt that MCMC methods using these kernels would have any use in any realistic scenarios due to the inefficient sampling. Also, I believe that extending the approach to more complex kernels and loss functions might be very difficult. Second, the assumption of unique zeros for the loss function is rather strong one. I'm not sure if it is any better than the strong convexity assumption in [1] and most certainly it is worse than assuming Lipschitzity as in [1].

Besides the technical limitations, I believe the paper is bit difficult to follow. There are many highly technical concepts that are rushed through which makes it difficult to connect the different parts of the paper.

Minor comments / typos:

- Line 59: Reyni
- Definition 1 is phrased bit oddly in my opinion. I understand what you mean, but for a reader unfamiliar with DP this might be bit difficult to follow. Maybe "for all mechanisms $\mu \in \mathcal{M}$"?
- Definition 2, B is not used anywhere after it is introduced.
- Theorem 1, might make sense to include a description of the V-norm.
-  Line 89: "common common"
- Should there be a minus in front of the $\log(n)$ in Eq. (2)?
- Definition 4., missing the cite?
- Page 4, missing closing parenthesis after "i.e. $\mathcal{Y}$ ..."
- Page 4, before you define $\tilde{f}_X(y)$, is the respectively correct? To me it now reads that (1-k) gets assigned to atom.
- Theorem 2, I think you should use X instead of x in $x \in \mathcal{X}^{n}$ to be more consistent with the notation.
- Theorem 2, claim 2, are the samples really from $f$? I would imagine they are from $\tilde{f}$ as described in the output of Alg. 1.
- Theorem 2, is the expected number of samples the same as expected time of return to the atom?
- Appendix, I don't quite follow the proof for Thm 2. How do you reach the expression after "This implies:"? How is it possible that there is still a Diracs delta (in $\tilde{\eta}$) remaining after the integral?
- Page 6, last equality. Where does this come from?
- Font in Figure 1 is way too small.
- I don't follow Figure 1 at all. What are the subtitles "n=100, n=200, n=500", the sample sizes? What are the axes? How come delta decrease as the chain lenght increases? The right column shows results on simulated data, and I presume the boxes show some kind of standard error. Are these results purely analytical, or are these errors computed over some simulation? If latter, what was the number of repeats/simulations?
- Figure 2, is the y-label "Err" correct? I think you mean utility.
- reference 16, the title of the paper is truncated
- reference 17, "HItomi"

References:

[1] Arun Ganesh and Kunal Talwar. Faster differentially private samplers via rényi divergence analysis of discretized langevin mcmc. In H. Larochelle, M. Ranzato, R. Hadsell, M. F. Balcan, and H. Lin, editors, Advances in Neural Information Processing Systems, volume 33, pages 7222–7233. Curran Associates, Inc., 2020.

**Time Spent Reviewing:**

5 hours

---

> ### Author Response · Authors · 2021-08-09
> **Response for reviewer DMjk**
>
> We sincerely appreciate you taking the time to write a thorough review with detailed minor comments. We also appreciate that you recognize the novelty and usefulness of our approach. We would like to address your high level concerns as well as some relevant minor comments.
>
> First, we agree that assuming $L_X$ has a unique minimum is quite restrictive. On second pass of our results, thankfully, we believe we do not need such an assumption! Because our proofs do not rely on a strict inequality in $L_X$, the confidential atom can be chosen as any point that minimizes $L_X$, regardless of its uniqueness. This will also require a slight modification in how to use samples from $\tilde{f}_X$ to sample from $f_X$, depending on whether or not $Y$ assigns mass to the set of outputs that minimize $L_X$. Regardless, we can and will modify our proofs to accommodate this change in the final version (as well as acknowledging you for pointing this out), eliminating this problem.
>
> Next, as alluded to in our responses to reviewer Dxmp and Jv9s, we choose our first example for two specific reasons. First, we wanted example chains for which both standard MCMC methods as well as our method have tight, analytically tractable convergence rates and expected runtimes, respectively. Second, we wanted to illustrate the counter-intuitive phenomena that occurs when controlling for the worst-case mixing time instead of that associated with the confidential data; our method avoids this problems. In practice, we certainly agree that these problems would not be implemented with our toy methods and would be much better suited to Langevin or Hamiltonian MC, for example. However, calculating the realized privacy guarantees for modern methods like these raises the same problems where the results are given as high-probability asymptotic bounds; furthermore, the results often depend on more stringent assumptions about $L_X$ like those in (Ganesh and Talwar, 2020). There is also, unfortunately, the unavoidable problem that comparing $\epsilon$-DP and $(\epsilon, \delta)$-DP or $(\epsilon, \alpha)$-RDP methods is difficult to do in an balanced, ``apples-to-apples" manner. Our proposed method may struggle with similar calculation details as those pointed out above, particularly for implementations where the minorizing constant may be difficult to characterize. However, this affects not just our work but almost all existing literature on perfect sampling. For this reason, we believe our methods are best suited to low-dimensional releases where data curators need provable guarantees, and we acknowledge that this won't meet the needs of data curators for whom a small $\delta$ or $\alpha$ cost does not concern them. There are many other reasons why we believe the low-dimensional use case is both practically and theoretically important; please see our comments in review Jv9s for further discussion.
>
> Selected responses to minor comments:
>
> - Typo in Eq.1: $-m$ should be $m$, as $r \in (0, 1)$ for both Eq. 1 and 2, so that the power in Eq.2 is $log(n)$
>
> - Theorem 2, claim 2: the samples are from $f_X$, not $\tilde{f}_X$- part of the runtime calculation includes an additional geometric factor due to using $\tilde{f}_X$ to sample $f_X$. This will be modified, as previously discussed, to account for non-uniqueness in $L_X$ minimum.
>
> - Theorem 2: the expected number of samples is the number of proposed samples needed in ALL steps of the algorithm, not just those dictating return to the atom.
>
> - Page 6: will add a citation here, this is a classical result from (Mengersen and Tweedie, 1996)
>
> - Figure 1. The left column shows the worst-case $\delta$ from two different MCMC implementations of the exponential mechanism, and the right column calculates the distribution (as boxplots) of our perfect sampler expect proposals analytical upper bound under different $X$ distributions. This should be fully specified in the paper text and appendix.

---

> > ### Comment · Reviewer_DMjk · 2021-09-10
> > **After the rebuttal**
> >
> > Thanks authors for addressing many of my concerns. I still have some concerns regarding the applicability of the method, but I will raise my score to weak accept for the novel theoretical contributions.

---

### Official Review · Reviewer_Jv9s · 2021-07-14

**Rating:** 5
**Confidence:** 4

**Summary:**

The paper proposes a MCMC approach to mimicking the DP Exponential Mechanism. Moreover, its main claim to fame is that it gives sufficient conditions for finite-time convergence. These conditions are based on the existence of "atoms" (Def.4), which, in this work's setting, is the optimal state. In addition, the approach is evaluated on two cases (mean estimation and Ridge Rigression on a bounded domain.

**Limitations And Societal Impact:**

Irrelevant.

**Main Review:**

I am a big fan of using MCMC to tackle infeasible problems, ones for which that Exponential Mechanism is too "time consuming." But this paper, I am afraid, does not really give useful bounds for such problems. Instead, it relies heavily on \beta = the infimum of the transition probability between ANY state in the space and the atom (Thm 2, bulletin 1); and \inf{p_accept} -- the expected mass entering any state. Now, if we examine Figure 1, LHS, we see an interesting phenomena: as \epsilon *increases* it takes longer for the chain to get close (in TV-distance) to the stationary distribution (the output distribution of the Exponential Mechanism). This is a direct result of the dependency on the inverse of \inf{p_accept} --- as \eps increases, the probability of transition into a point which is very unlikely to be a solution decreases exponentially, causing the runtime to be longer. But the result is that in many cases -- especially over large state space -- the convergence bound given in this work aren't really useful...

What this paper does show, is that certain cases can be implemented efficiently. I am, however, a bit on the fence as to how important are these cases. On the one hand, I am very much in favor showing any result one can about the convergence of MCMC into the Exp-Mech, as it is so useful in real life applications; on the other hand, I am not convinced that there's enough "substance" here to be of any true interest to the DP community. I eagerly await the authors' rebuttal and the following discussion to make my final decision, but, as it currently stands, I do not think these results are important enough...

**Time Spent Reviewing:**

Refuse to time myself

---

> ### Author Response · Authors · 2021-08-09
> **Response for reviewer Jv9s**
>
> Thank you for your review. To start, we'd like to clarify some claims made in your summary and review. First, our results do not ``mimic" the exponential mechanism but instead exactly implements it, which standard finite-chain MCMC does not. Our main result, Theorem 2, is an exact implementation of the exponential mechanism. The corollary to this result, Corollary 1, generates a sample from a distribution that is very close to the original target distribution of the exponential mechanism; however, the sample is still $\epsilon$-DP. In both cases, the result is $\epsilon$-DP, which is not true for any MCMC scheme where the target distribution is only reached asymptotically.
>
> Second, while our perfect sampling approach is partially based on MCMC, techniques to analyze standard MCMC convergence cannot be used to analyze perfect sampling runtimes. Markov Chains are run to a single, finite length, at which point one compares the distribution of this finite length chain to the target. In contrast, a perfect sampler's output is always the target distrubtion; however, the number of total proposals needed to generate this sample, $N_{\mathrm{prop}}$ in our notation, has its own distribution.
>
> Third, the recurrent atoms used in instantiating Theorem 2 need to be measurable sets for the distribution on $Y$, and most non-discrete distributions do not naturally admit these atoms. This is why we discuss modifying the base measure and base distribution of our target to generically accommodate this criteria, a crucial ingredient for expanding the applicability of our results.
>
> Next, we'd like to address your concerns about the usefulness of the bounds derived in Theorem 2 and Corollary 1. You are correct in stating that the minorization constant $\beta$ does ``heavy lifting" in both MCMC and perfect sampling regimes. However, to clarify, the minorization constant serves very different purposes in each case. For standard MCMC implementations, $\beta$ appears in the expression to bound total variation distance for a finite chain, as in $V$-geometric ergodicity for example. For perfect sampling, though, $\beta$ appears in the effective runtime of the algorithm as a parameter in the random length of an inner loop used in determining the total number of proposals needed for the entire implementation. In both cases, though, heavy dependence on the minorization constant is a defining feature not just of our work but of existing ergodic theory and perfect sampling literature. In particular, (Lee et al, 2014) derive some converse results that show that without the minorization constant, perfect sampling using their approach is not possible with finite expected runtime. Therefore any limitations concerning the realized values of $\beta$ are no different than those in existing literature, and further emphasize the importance of recognizing the tradeoff between runtime and privacy guarantees.
>
> As you correctly noted, the convergence bound suffers from a curse of dimensionality problem. However, we do not believe this is a substantial demerit against our work, and we'd like to explain why. First, we are transparent in our belief that this work is best suited to lower-dimensional, non-discrete instances of the exponential mechanism. This case arises frequently in the social, behavioral, and biomedical sciences, of which there are numerous data curators who want to share statistical results using these mechanisms without concerns for scale. Furthermore, to the best of our knowledge, we could not find existing literature that addresses this common use case. Second, these dimensionality problems plague the utility of ANY implementation of the exponential mechanism itself, as the sensitivity of the loss function increases as the dimension of the output space increases. Dimension reduction is an essential component of effective practical usage for the exponential mechanism, regardless of the implementation barrier we are trying to overcome. Third, the bounds we derive in Theorem 2 and Corollary 1 are bounds on the expectation of $N_{\mathrm{prop}}$, the total number of proposed samples needed to generate the single perfect sample. $N_{\mathrm{prop}}$ is a random variable with a complex distribution, and the weight of its tail can vary substantially depending on how efficiently the perfect sampler is implemented. Regardless, though, the algorithm will provably converge in finite time, and bounding $E[N_{\mathrm{prop}}]$ is only one way to represent this property; it cannot and should not be interpreted in the same way as a convergence bound or rate that one would see in analyzing a Markov Chain that only converges as the chain length becomes infinite.
>
> We believe these results are important because they address a real barrier to wider adoption of DP using a new theoretical toolkit. We also acknowledge that the kinds of results we are presenting here are different than what has traditionally been ``interesting" to the DP community, which is usually asymptotic sample complexity results based on applying learning theory techniques to private estimation procedures. Our approach borrows from a different set of theoretical tools, but requires explicit definitions of asymptotic constants usually ignored by theoretical DP researchers. For example, the results of (Ganesh and Talwar, 2020) are certainly interesting and yield nice rates of convergence and dependence on dimension; however, they are not helpful to practitioners trying to ensure that a Langevin dynamics implementation of the exponential mechanism is $(\epsilon, \delta)$-DP for known constants $\epsilon$ and $\delta$. Moreover, they offer no implementation guidance as even the Langevin step size is asymptotic; this is especially important if the implementation is in high-dimensions.
>
> We sincerely hope that this paper will pique the interest of DP researchers to apply and extend a new set of theoretical tools to address a real implementation problem under different logistical constraints. We authors have worked on DP from both theoretical and applied perspectives. In general, we've noticed that data curators using DP frequently express concerns that implementation barriers are not taken seriously by theoretical DP researchers. We hope that by drawing connections between this line of work and a new mathematical toolkit, we can motivate theoretical DP researches not only to take this problem seriously, but do so in a mathematically satisfying way, making everyone the better.

---

### Official Review · Reviewer_UUA7 · 2021-07-16

**Rating:** 7
**Confidence:** 2

**Summary:**

Since exponential mechanism, which is important in differential privacy (DP), is usually difficult to sample  because of intractable distribution and existing sampling methods like MCMC cannot guarantee privacy, this work focuses on implementable finite sampling method for exponential mechanism based on ergodic theory with include artificial atoms into target distribution to balance tradeoffs by guaranteeing privacy under DP manner and improving runtime compared with previous methods.

**Limitations And Societal Impact:**

No potential negative societal impact has seen.

**Main Review:**

Pros:
1. Although this work is based on MCMC, this work guarantees \epsilon-DP by using atomic regeneration to introduce an exact finite-runtime algorithm to implement exponential mechanism. And even the modified version can outputs a sample with smaller utility and shorter runtime as well as guarantee for \epsilon-DP, which are crucial tradeoffs.
3.  This work provides rigorous definitions and theorems to theoretically explain the effectiveness and correctness of proposed algorithms.
4. This work carefully states the limitation of the proposed approach and provide potential improvement, such as generic utility for the exponential mechanism.

Cons:
1. Be honest, I work on MPC side and has bare knowledge of DP, so, as far as I am concerned, there is no obivious cons in this work and I am willing to see other reviewers’ comments to make the next move.

**Time Spent Reviewing:**

3

---

> ### Author Response · Authors · 2021-08-09
> **Response for reviewer UUA7**
>
> Thank you for your positive review and for understanding the contributions, limitations, and importance of our work! We'd like to emphasize that we intend these methods be used for lower dimensional releases from the exponential mechanism, as higher-dimensional releases tend to suffer from curse-of-dimensionality effects in output utility due to the sensitivity of the underlying space. However, lower dimensional releases, particularly when $\mathcal{Y}$ is not discrete, are used very frequently for statistical analyses in the social, behavioral, and biomedical sciences. Therefore we argue that despite these limitations, we still offer methods of value to practitioners hoping to implement differential privacy while overcoming the barrier of intractable target densities. See our response to reviewer Jv9s for more detail.

---

### Official Review · Reviewer_Dxmp · 2021-07-16

**Rating:** 6
**Confidence:** 2

**Summary:**

This paper focuses an exact finite runtime sampling algorithm for the exponential mechanism by including an artificial atom in the target distribution.

The main contributions are:

1. The paper proposes an exact finite-runtime algorithm (ConfAtomPerfect) for implementing the exponential mechanism with ε-DP guarantees using atomic regeneration.

2. The paper derives a modification of the previous algorithm (RandomAtomPerfect) that satisfies ε-DP but outputs a sample from a modified target distribution with smaller utility and shorter runtime.

**Limitations And Societal Impact:**

Yes

**Main Review:**

Comments:

1. Lack of experiments. The paper only contains 2 toy examples. Moreover, it also lacks comparative experiments with other related methods. So, it is difficult to evaluate the significance of the contribution.

2. There are too few references for similar/related methods.

**Time Spent Reviewing:**

5

---

> ### Author Response · Authors · 2021-08-09
> **Response for reviewer Dxmp**
>
> Thank you for your review and for recognizing our contributions. We'd like to acknowledge an additional contribution of our paper: our example demonstrates that while the guarantees provided by MCMC $(\epsilon, \delta)$-DP implementations depend on the slowest possible mixing chain, the guarantees in our methods depend only on the existing realization of the confidential data.
>
> We would like to address your concerns about the lack of experiments. The first major difficulty is that releases which satisfy different DP formalisms are not directly comparable in terms of runtime or utility. Although there are hierarchical relationships between $(\epsilon, \delta)$-DP, $(\epsilon, \alpha)$-RDP, and $\epsilon$-DP, there is no even playing field with which one can determine if one method is ``state of the art" compared to another. The second major difficulty is that, for the methods we would ideally compare our work to, the privacy guarantees are only known as asymptotic rates. For a fair comparison, we would need to calculate and upper bound any constants in convergence rate arguments to compare $(\epsilon, \delta)$-DP and $\epsilon$-DP releases. This would make it difficult to disentangle the effects of the algorithm itself with the way we calculate an upper bound and however tight or loose it is. This also makes it difficult to compare our work against more modern MCMC methods like Hamiltonian MCMC or Langevin dynamics; see our response to reviewer Dmjk for further discussion. Additionally, trying to solve this problem numerically is computationally infeasible, since calculating Total Variation (TV) distance generically can be a computationally expensive task, even before attempting to find the slowest mixing chain numerically for an arbitrary database.
>
> We chose examples for which we could analytically calculate tight constants using well-known classical results in order to illustrate the effects of sample size, $\epsilon$, and proposal distribution on runtime. Even this comparison is not apples-to-apples, as a Markov chain's length is deterministic, and the total number of proposals needed to implement the perfect sampling algorithm is random. However, this comparison demonstrates the phenomena we describe theoretically in our paper, which is why we chose to include it in our paper. This does not demonstrate a limitation of where our results could apply; instead, it demonstrates a limitation of standard MCMC implementations of the exponential mechanism, for which the $\delta$ cost incurred by finite chains is difficult to calculate in practice.
>
> If accepted, and if space permits within the page limit, we will include an experiment in which we show the distribution of perfect sampling runtimes for a particular instance of the exponential mechanism while varying $\epsilon$. This would better convey the probabilistic nature of the runtime and the effect of privacy budget on the results. Additionally, this could be used to demonstrate scenarios in which our computed upper bound is tight or loose; the total number of proposed samples is the composition of runtime effects from multiple sources (as enumerated in the proof and proof sketch), so the actual distribution of $N_{\mathrm{prop}}$ is quite complex.  Because of the same points above, though, we could not reasonably compare this to existing methods in a manner similar to standard ML papers where methods are compared on a level playing field. These examples are meant to illustrate the differences that arise under different DP formalisms and parameter configuration as they relate to our theoretical results.
>
> Next, we want to address your concerns about the ``lack of references for similar / related methods." To the best of our knowledge, we could not find existing literature on methods that guaranteed exact $\epsilon$-DP implementations, instead finding works like (Ganesh and Talwar, 2020) that rely on rates of convergence in a weaker privacy definition. Given our experience with existing privacy literature, we do not find this particularly surprising. First, our paper addresses a real implementation issue for a well-known privacy mechanism, but implementation barriers are usually not given such a mathematical treatment in the privacy literature. Second, most DP papers rely heavily on asymptotic results from statistical learning theory and information theory in their arguments; not only do we require finite sample results, but we require any constants within these results to be calculable. We discuss this more in our response to reviewer Jv9s. Reviewer Dmjk noted one classical result missing a reference in their minor comments, which we will add. If there are any particular references you believe are still missing, please let us know and we would be happy to update the paper.

---

### Official Review · Reviewer_Dir6 · 2021-08-03

**Rating:** 6
**Confidence:** 2

**Summary:**

The paper studies sampling algorithms for the exponential mechanism by using ergodic theory and perfect simulation. They also give an algorithm that provides pure DP at the loss of some utility.

**Limitations And Societal Impact:**

I am not entirely sure if the current implementation does not suffer with floating point arthematic. The two examples are interesting, but seems little restrictive.

**Main Review:**

The paper has three contributions: (i) an exact algorithm that runs in a finite time and provides approximate DP, (ii) an exact finite time algorithm for pure DP, and (iii) an algorithm that has a faster run time but at an expense of loss of utility.

The paper makes an assumption that the transition kernel satisfies a minorization property. As an example, they show efficient sampling for two cases: (i) d-dimensional mean with L_1 loss, where they compare pure and approx DP cases, and (ii) ridge regression K-norm gradient mechanism.

I think the paper is well written and I was able to follow the first 9 pages of the paper. I could not verify the proof in the time, so I cannot vouch for the correctness of the paper. Under the assumption that the proof is fine, there is definite merit.

**Time Spent Reviewing:**

4

---

> ### Author Response · Authors · 2021-08-09
> **Response for reviewer Dir6**
>
> Thank you for your review and for acknowledging contributions (ii) and (iii) that you've listed in your review. Note that, in your ordering, (i) is the current standard approach that provides approximate DP, while our approach also guarantees pure DP. We'd like to emphasize one additional contribution: our example demonstrates that while the guarantees provided by (i) depend on the slowest possible mixing chain, the guarantees in (ii) and (iii) depend only on the existing realization of the confidential data.
>
> Next, we'd like to briefly comment on the limitations you noted in your review. We agree that floating point and other side channel vulnerabilities are an issue not just with our proposed method, but any sampling-based method when the output space of the exponential mechanism is not discrete. This has been discussed before in the DP literature, and similar papers such as Ganesh and Talwar (2020) acknowledge the same issue.
>
> Next, we chose simple examples so that the rates of convergence for the slowest mixing chains across all possible databases were analytically tractable; this allowed us to compare existing approaches based on asymptotic convergence of MCMC to our perfect sampling approach, noting that the former is $(\epsilon,\delta)$ and the latter is $\epsilon$-DP. We want to strongly emphasize that this is not a limitation of our perfect sampling method to more complex instances of the exponential mechanism. Instead, it is a limitation of the examples for which we could do this comparison and reasonably bound the $\delta$ incurred from the slowest mixing chain across all possible databases. We discuss this further in the response to reviewer Dmjk, second paragraph. These examples are meant to illustrate the phenomena we described theoretically in our paper, specifically the benefit of having the implementation algorithm depend on the confidential data.

---

### Decision · Program_Chairs · 2021-09-27

**Decision:**

Accept (Poster)

**Comment:**

The reviewers were overall positive about the paper, modulo the following concerns, which came up during the discussion period.

1. Comparison to prior work, especially to Ganesh and Talwar. In particular, the reviwers will be more satisfied if even just empirically -- one shows that the algorithm outperforms some existing algorithms on some measure (say, runtime), and
2. Since the assumptions in the paper are different than prior work, it would be important to contrast them with prior work. In particular, w.r.t. curse of dimensionality.